# BNN+: Improved Binary Network Training

## ABSTRACT

Deep neural networks (DNN) are widely used in many applications. However, their deployment on edge devices has been difficult because they are resource hungry. Binary neural networks (BNN) help to alleviate the prohibitive resource requirements of DNN, where both activations and weights are limited to 1-bit. We propose an improved binary training method (BNN+), by introducing a regularization function that encourages training weights around binary values. In addition to this, to enhance model performance we add trainable scaling factors to our regularization functions. Furthermore, we use an improved approximation of the derivative of the sign activation function in the backward computation. These additions are based on linear operations that are easily implementable into the binary training framework. We show experimental results on CIFAR-10 obtaining an accuracy of $86.5\%$, on AlexNet and $91.3\%$ with VGG network. On ImageNet, our method also outperforms the traditional BNN method and XNOR-net, using AlexNet by a margin of $4\%$ and $2\%$ top-1 accuracy respectively.

## 1 INTRODUCTION

Deep neural networks (DNNs) have demonstrated success for many supervised learning tasks ranging from voice recognition to object detection (Szegedy et al., 2015; Simonyan & Zisserman, 2014; Iandola et al., 2016). The focus has been on increasing accuracy, in particular for image tasks, where deep convolutional neural networks (CNNs) are widely used. However, their increasing complexity poses a new challenge, and has become an impediment to widespread deployment in many applications; specifically when trying to deploy such models to resource constrained and lower-power devices. A typical DNN architecture contains tens to thousands of layers, resulting in millions of parameters. As an example, AlexNet (Krizhevsky et al., 2012) requires 200MB of memory, VGG-Net (Simonyan & Zisserman, 2014) requires 500MB memory. Large model sizes are further exacerbated by their computational cost, requiring GPU implementation to allow real-time inference. Such requirements evidently cannot be accustomed by edge devices as they have limited memory, computation power, and battery. This motivated the community to investigate methods for compressing and reducing computation cost of DNNs.

To make DNNs compatible with the resource constraints of low power devices, there have been several approaches developed, such as network pruning (LeCun et al., 1990), architecture design (Sandler et al., 2018), and quantization (Courbariaux et al., 2015; Han et al., 2015). In particular, weight compression using quantization can achieve very large savings in memory, where binary (1-bit), and ternary (2-bit) approaches have been shown to obtain competitive accuracy (Hubara et al., 2016; Zhu et al., 2016; Tang et al., 2017). Using such schemes reduces model sizes by 8x to 32x depending on the bit resolution used for computations. In addition to this, the speed by quantizing the activation layers. In this way, both the weights and activations are quantized so that one can replace the expensive dot products and activation function evaluations with bitwise operations. This reduction in bit-width benefits hardware accelerators such as FPGAs and neural network chips.

An issue with using low-bit DNNs is the drastic drop in accuracy compared to its full precision counterpart, and this is made even more severe upon quantizing the activations. This problem is largely due to noise and lack of precision in the training objective of the networks during back-propagation (Lin et al., 2017). Although, quantizing the weights and activations have been attracting large interests thanks to their computational benefits, closing the gap in accuracy between the full precision and the quantized version remains a challenge. Indeed, quantizing weights cause drastic information loss and make neural networks harder to train due to a large number of sign fluctuations in the

weights. Therefore, how to control the stability of this training procedure is of high importance. In theory, it is infeasible to back-propagate in a quantized setting as the weights and activations employed are discontinuous and discrete. Instead, heuristics and approximations are proposed to match the forward and backward passes. Often weights at different layers of DNNs follow a certain structure. How to quantize the weights locally, and maintaining a global structure to minimize a common cost function is important (Li et al., 2017).

Our contribution consists of three ideas that can be easily implemented in the binary training framework presented by Hubara et al. (2016) to improve convergence and generalization accuracy of binary networks. First, the activation function is modified to better approximate the sign function in the backward pass, second we propose two regularization functions that encourage training weights around binary values, and lastly a scaling factor is introduced in both the regularization term as well as network building blocks to mitigate accuracy drop due to hard binarization. Our method is evaluated on CIFAR-10 and ImageNet datasets and compared to other binary methods. We show accuracy gains to traditional binary training.

## 2 RELATED WORK

We focus on challenges present in training binary networks. The training procedure emulates binary operations by restricting the weights and activations to single-bit so that computations of neural networks can be implemented using arithmetic logic units (ALU) using XNOR and popcount operations. More specifically, XNOR and popcount instructions are readily available on most CPU and GPU processing units. Custom hardware would have to be implemented to take advantage of operations with higher bits such as 2 to 4 bits. The goal of this binary training is to reduce the model size and gain inference speedups without performance degradation.

Primary work done by Courbariaux et al. (2015) (BinaryConnect) trains deep neural networks with binary weights $\{-1, +1\}$. They propose to quantize real values using the sign function. The propagated gradient applies update to weights $|w| \leq 1$. Once the weights are outside of this region they are no longer updated, this is done by clipping weights between $\{-1, +1\}$. In that work, they did not consider binarizing the activation functions. BNN (Hubara et al., 2016) is the first purely binary network quantizing both the weights and activations. They achieve comparable accuracy to their prior work on BinaryConnect, and achieve significantly close performance to full-precision, by using large and deep networks. Although, they performed poorly on large datasets like ImageNet (Russakovsky et al., 2015). The resulting network presented in their work obtains $32\times$ compression rate and approximately $7\times$ increase in inference speed.

To alleviate the accuracy drop of BNN on larger datasets, Rastegari et al. (2016) proposed XNOR-Net, where they strike a trade-off between compression and accuracy through the use of scaling factors for both weights and activation functions. In their work, they show performance gains compared to BNN on ImageNet classification. The scaling factors for both the weights and activations are computed dynamically, which slows down training performance. Further, they introduced an additional complexity in implementing the convolution operations on the hardware, slightly reducing compression rate and speed up gains. DoReFa-Net (Zhou et al., 2016) further improves XNOR-Net by approximating the activations with more bits. The proposed rounding mechanism allows for low bit back-propagation as well. Although they perform multi-bit quantization, their model still suffers from large accuracy drop upon quantizing the last layer.

Later in ABC-Net, Tang et al. (2017) propose several strategies, adjusting the learning rate for larger datasets. They show BNN achieve similar accuracy as XNOR-Net without the scaling overhead by adding a regularizer term which allows binary networks to generalize better. They also suggest a modified BNN, where they adopted the strategy of increasing the number of filters, to compensate for accuracy loss similar to wide reduced-precision networks (Mishra et al., 2017). More recently, Liu et al. (2018) developed a second-order approximation to the sign activation function for a more accurate backward update. In addition to this, they pre-train the network in which they want to binarize in full precision using the hard tangent hyperbolic (htanh) activation, see Figure 2. They use the pre-trained network weights as an initialization for the binary network to obtain state of the art performance.

## 3 IMPROVED BINARY TRAINING

Training a binary neural network faces two major challenges: on weights, and on activation functions. As both weights and activations are binary, the traditional continuous optimization methods such as SGD cannot be directly applied. Instead, a continuous approximation is used for the sign activation during the backward pass. Further, the gradient of the loss with respect to the weights are small. So as training progresses weight sign remains unchanged. These are both addressed in our proposed method. In this section, we present our approach to training 1-bit CNNs in detail.

### 3.1 BINARY TRAINING

We quickly revisit quantization through binary training as first presented by (Courbariaux et al., 2015). In (Hubara et al., 2016), the weights are quantized by using the sign function which is $+1$ if $w > 0$ and $-1$ otherwise.

In the forward pass, the real-valued weights are binarized to $w^b$, and the resulting loss is computed using binary weights throughout the network. For hidden units, the sign function non-linearity is used to obtain binary activations. Prior to binarizing, the real weights are stored in a temporary variable $w$. The variables $w$ are stored because one cannot back-propagate through the sign operation as its gradient is zero everywhere, and hence disturbs learning. To alleviate this problem the authors suggest using a straight through estimator (Hinton, 2012) for the gradient of the sign function. This method is a heuristic way of approximating the gradient of a neuron,

$$\frac{dL(w)}{dw} \approx \frac{dL}{dw}\bigg|_{w=w^b} \mathbf{1}_{\{|w|\leq 1\}} \tag{1}$$

where $L$ is the loss function and $\mathbf{1}(.)$ is the indicator function. The gradients in the backward pass are then applied to weights that are within $[-1, +1]$. The training process is summarized in Figure 1. As weights undergo gradient updates, they are eventually pushed out of the center region and instead make two modes, one at $-1$ and another at $+1$. This progression is also shown in Figure 1.

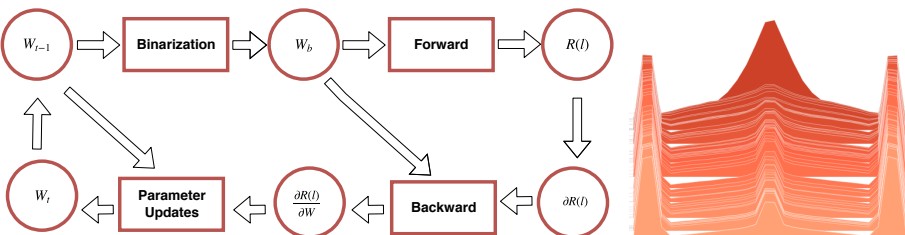

Figure 1: Binary training, where arrows indicate operands flowing into operation or block. Reproduced from Guo (2018) (left). A convolutional layer depicting weight histogram progression during the popular binary training. The initial weight distribution is a standard Gaussian (right).

### 3.2 IMPROVED TRAINING METHOD

Our first modification is on closing the discrepancy between the forward pass and backward pass. Originally, the sign derivative is approximated using the $\mathrm{htanh}(x)$ activation, as in Figure 2. Instead, we modify the Swish-like activation (Ramachandran et al., 2017; Elfwing et al., 2018; Hendrycks & Gimpel, 2016), where it has shown to outperform other activations on various tasks. The modifications are performed by taking its derivative and centering it around 0

$$\mathrm{SS}_\beta(x) = 2\sigma(\beta x)\left[1 + \beta x\{1 - \sigma(\beta x)\}\right] - 1, \tag{2}$$

where $\sigma(z)$ is the sigmoid function and the scale $\beta > 0$ controls how fast the activation function asymptotes to $-1$ and $+1$. The $\beta$ parameter can be learned by the network or be hand-tuned as a hyperparameter. As opposed to the Swish function, where it is unbounded on the right side, the

modification makes it bounded and a valid approximator of the sign function. As a result, we call this activation SignSwish, and its gradient is

$$\frac{d\mathrm{SS}_\beta(x)}{dx} = \frac{\beta\{2 - \beta x \tanh(\frac{\beta x}{2})\}}{1 + \cosh(\beta x)} \tag{3}$$

which is a closer approximation function compared to the htanh activation. Comparisons are made in Figure 2.

Hubara et al. (2016) noted that the STE fails to learn weights near the borders of $-1$ and $+1$. As depicted in Figure 2, our proposed SignSwish activation alleviates this, as it remains differentiable near $-1$ and $+1$ allowing weights to change signs during training if necessary.

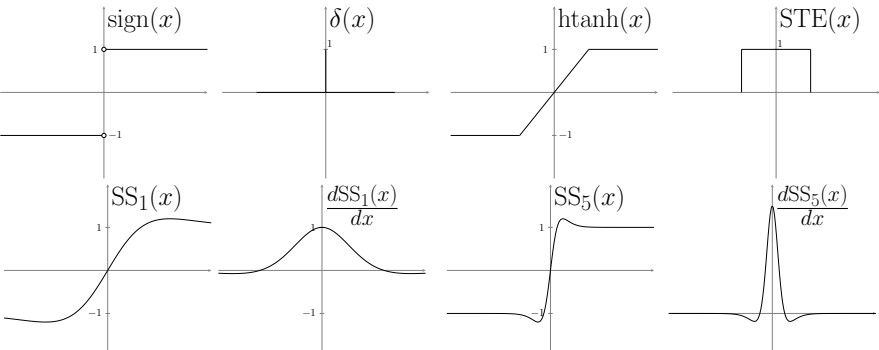

Figure 2: Forward and backward approximations. (Top Left) The true forward and backward functions. (Top Right) BNN training approximation. (Bottom Left) Swish function and its derivative. (Bottom Right) The modified activation, in this case $SS(x)$ is plotted for $\beta = 5$.

Note that the derivative $\frac{d}{dx}\mathrm{SS}_\beta(x)$ is zero at two points, controlled by $\beta$. Indeed, it is simple to show that the derivative is zero for $x \approx \pm 2.4/\beta$. By adjusting this parameter beta, it is possible to adjust the location at which the gradients start saturating. In contrast to the STE estimators, where it is fixed. Thus, the larger $\beta$ is, the closer the approximation is to the derivative of the sign function.

### 3.2.1 REGULARIZATION FUNCTION

In general, a regularization term is added to a model to prevent over-fitting and to obtain robust generalization. The two most commonly used regularization terms are $L_1$ and $L_2$ norms. If one were to embed these regularization functions in binary training, it would encourage the weights to be near zero; though this does not align with the objective of a binary network. Instead, it is important to define a function that encourages the weights around $-1$ and $+1$. Further, in Rastegari et al. (2016) they present a scale to enhance the performance of binary networks. This scale is computed dynamically during training, using the statistics of the weights. To make the regularization term more general we introduce scaling factors $\alpha$, resulting in a symmetric regularization function with two minimums, one at $-\alpha$ and another at $+\alpha$. As these scales are introduced in the regularization function and are embedded into the layers of the network they can be learned using backpropagation.

The Manhattan regularization function is defined as

$$R_1(w) = |\alpha - |w||, \tag{4}$$

whereas the Euclidean version is defined as

$$R_2(w) = (\alpha - |w|)^2, \tag{5}$$

where $\alpha > 0$ is the scaling factor. As depicted in Figure 3, in the case of $\alpha = 1$ the weights are penalized at varying degrees upon moving away from the objective quantization values, in this case, $\{-1, +1\}$.

The proposed regularizing terms are inline with the wisdom of the regularization function $R(w) = (1 - w^2)\mathbf{1}_{\{|w| \leq 1\}}$ as introduced in Tang et al. (2017). A primary difference are in introducing a trainable scaling factor, and formulating it such that the gradients capture appropriate sign updates to the weights. Further, the regularization introduced in Tang et al. (2017) does not penalize weights that are outside of $[-1, +1]$. One can re-define their function to include a scaling factor as $R(w) = (\alpha - w^2)\mathbf{1}_{\{|w| \leq \alpha\}}$. In Figure 3, we depict the different regularization terms to help with intuition.

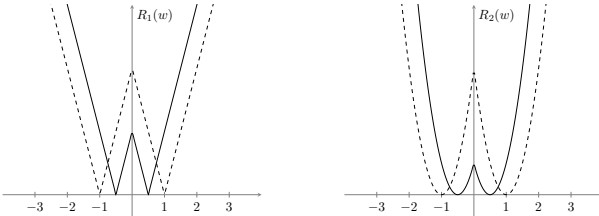

Figure 3: $R_1(w)$ (left) and $R_2(w)$ (right) regularization functions for $\alpha = 0.5$ (solid line) and $\alpha = 1$ (dashed line). The scaling factor $\alpha$ is trainable, as a result the regularization functions can adapt accordingly.

### 3.3 BNN+: Training Method

Combining both the regularization and activation ideas, we modify the training procedure by replacing the sign backward approximation with that of the derivative of $\text{SS}_\beta$ activation (2). During training, the real weights are no longer clipped as in BNN training, as the network can back-propagate through the $\text{SS}_\beta$ activation and update the weights correspondingly.

Additional scales are introduced to the network, which multiplies into the weights of the layers. The regularization terms introduced are then added to the total loss function,

$$J(W, b) = L(W, b) + \lambda \sum_l R(W_l, \alpha_l) \tag{6}$$

where $L(W, b)$ is the cost function, $W$ and $b$ are the sets of all weights and biases in the network, $W_l$ is the set weights at layer $l$ and $\alpha_l$ is the corresponding scaling factor. Here, $R(.)$ is the regularization function (4) or (5). Further, $\lambda$ controls the effect of the regularization term. To introduce meaningful scales, they are added to the basic blocks composing a typical convolutional neural network. For example, for convolutions, the scale is multiplied into the quantized weights prior to the convolution operation. Similarly, in a linear layer, the scales are multiplied into the quantized weights prior to the dot product operation. This is made more clear in the training algorithm 1.

The scale $\alpha$ is a single scalar per layer, or as proposed in Rastegari et al. (2016) is a scalar for each filter in a convolutional layer. For example, given a CNN block with weight dimensionality $(C_{\text{in}}, C_{\text{out}}, H, W)$, where $C_{\text{in}}$ is the number of input channels, $C_{\text{out}}$ is the number of output channels, and $H, W$, the height and width of the filter respectively, then the scale parameter would be a vector of dimension $C_{\text{out}}$, that factors into each filter.

As the scales are learned jointly with the network through backpropagation, it is important to initialize them appropriately. In the case of the Manhattan penalizing term (4), given a scale factor $\alpha$ and weight filter then the objective is to minimize

$$\min_\alpha \sum_{h,w} |\alpha - |W_{h,w}|| \tag{7}$$

The minimum of the above is obtained when

$$\alpha^* = \text{median}(|W|) \tag{8}$$

Similarly, in the case of the Euclidean penalty (5) the minimum is obtained when

$$\alpha^* = \text{mean}(|W|) \tag{9}$$

The scales are initialized with the corresponding optimal values after weights have been initialized first. One may notice the similarity of these optimal values with that defined by Rastegari et al. (2016), whereas in their case the optimal value for the weight filters and activations better matches the $R_2(w)$ goal. A difference is on how these approximations are computed, in our case they are updated on the backward pass, as opposed to computing the values dynamically.

The final resulting BNN+ training method is defined in Algorithm 1. In the following section, we present our experimental results and important training details.

---

**Algorithm 1** BNN+ training. $L$ is the unregularized loss function. $\lambda$ and $R_1$ are the regularization terms we introduced. $\text{SS}_\beta$ is the SignSwish function we introduced and $(\text{SS}_\beta)'$ is its derivative. $N$ is the number of layers. $\circ$ indicates element-wise multiplication. BatchNorm() specifies how to batch-normalize the activation and BackBatchNorm() how to back-propagate through the normalization. ADAM() specifies how to update the parameters when their gradients are known.

---

**Require:** a minibatch of inputs and targets $(x_0, x^*)$, previous weights $W$, previous weights' scaling factors $\alpha$, and previous BatchNorm parameters $\theta$.

**Ensure:** updated weights $W^{t+1}$, updated weights' scaling factors $\alpha^{t+1}$ and updated BatchNorm parameters $\theta^{t+1}$.

  {1. Forward propagation:}
  $s_0 \leftarrow x_0 W_0$ {We do not quantize the first layer.}
  $x_1 \leftarrow \text{BatchNorm}(s_0, \theta_0)$
  **for** $k = 1$ to $N - 1$ **do**
    $x_k^b \leftarrow \text{sign}(x_k)$
    $W_k^b \leftarrow \text{sign}(W_k)$
    $s_k \leftarrow \alpha_k x_k^b W_k^b$ {This step can be done using mostly bitwise operations.}
    $x_{k+1} \leftarrow \text{BatchNorm}(s_k, \theta_k)$
  **end for**

  {2. Backward propagation:}
  Compute $g_{x_N} = \frac{\partial L}{\partial x_N}$ knowing $x_N$ and $x^*$
  **for** $k = N - 1$ to $1$ **do**
    $(g_{s_k}, g_{\theta_k}) \leftarrow \text{BackBatchNorm}(g_{x_{k+1}}, s_k, \theta_k)$
    $g_{\alpha_k} \leftarrow g_{s_k} x_k^b W_k^b + \lambda \frac{\partial R_1}{\partial \alpha_k}$
    $g_{W_k^b} \leftarrow g_{s_k}^\top \alpha_k x_k^b$
    $g_{x_k^b} \leftarrow g_{s_k} \alpha_k W_k^b$
    {We use our modified straight-through estimator to back-propagate through $\text{sign}$:}
    $g_{W_k} \leftarrow g_{W_k^b} \circ (\text{SS}_\beta)'(W_k) + \lambda \frac{\partial R_1}{\partial W_k}$
    $g_{x_k} \leftarrow g_{x_k^b} \circ (\text{SS}_\beta)'(x_k)$
  **end for**
  $(g_{s_0}, g_{\theta_0}) \leftarrow \text{BackBatchNorm}(g_{x_1}, s_0, \theta_0)$
  $g_{W_0} \leftarrow g_{s_0} x_0$ {We did not quantize the first layer.}

  {3. The update:}
  **for** $k = 0$ to $N - 1$ **do**
    $\theta_k^{t+1}, W_k^{t+1}, \alpha_k^{t+1} \leftarrow \text{ADAM}(\eta, \theta_k, W_k, \alpha_k, g_{\theta_k}, g_{W_k}, g_{\alpha_k})$
  **end for**

---

## 4 EXPERIMENTAL RESULTS

We evaluate our proposed method with the accuracy performance of training using BNN+ scheme versus other proposed binary networks, Hubara et al. (2016); Rastegari et al. (2016); Tang et al. (2017). We run our method on CIFAR-10 and ImageNet datasets and show accuracy gains. They are discussed in their respective sections below.

## 4.1 CIFAR-10

The CIFAR-10 data (Krizhevsky & Hinton, 2009) consists of 50,000 train images and a test set of 10,000. For pre-processing the images are padded by 4 pixels on each side and a random crop is taken. We train both, AlexNet (Krizhevsky et al., 2012), and VGG (Simonyan & Zisserman, 2014) using the ADAM (Kingma & Ba, 2014) optimizer. The architecture used for VGG is $\text{conv}(256) \rightarrow \text{conv}(256) \rightarrow \text{conv}(512) \rightarrow \text{conv}(512) \rightarrow \text{conv}(1024) \rightarrow \text{conv}(1024) \rightarrow \text{fc}(1024) \rightarrow \text{fc}(1024)$ where $\text{conv}(\cdot)$ is a convolutional layer, and $\text{fc}(\cdot)$ is a fully connected layer. The standard $3 \times 3$ filters are used in each layer. We also add a batch normalization layer (Ioffe & Szegedy, 2015) prior to activation. For AlexNet, the architecture from Krizhevsky (2014) is used, and batch normalization layers are added prior to activations. We use a batch size of 256 for training. Many learning rates were experimented with such as $0.1, 0.03, 0.001$, etc, and the initial learning rate for AlexNet was set to $10^{-3}$, and $3 \times 10^{-3}$ for VGG. The learning rates are correspondingly reduced by a factor 10 every 10 epoch for 50 epochs. We set the regularization parameter $\lambda$ to $10^{-6}$, and use the regularization term as defined in (4). In these experiments weights are initialized using Glorot & Bengio (2010) initialization. Further, the scales are introduced for each convolution filter, and are initialized by sorting the absolute values of the weights for each filter and choosing the $75^{\text{th}}$ percentile value. The results are summarized in Table 1.

Table 1: Top-1 and top-5 accuracies (in percentage) on CIFAR-10, using Manhattan regularization function (4) with AlexNet and VGG.

|         |        | BNN+    | Full-Precision |
|---------|--------|---------|----------------|
| AlexNet | Top-1  | 86.49%  | 88.58%         |
|         | Top-5  | 98.92%  | 99.73%         |
| VGG     | Top-1  | 91.31%  | 90.89%         |
|         | Top-5  | 99.09%  | 99.76%         |

## 4.2 IMAGENET

The ILSVRC-2012 dataset consists of $\sim 1.2\text{M}$ training images, and 1000 classes. For pre-processing the dataset we follow the typical augmentation: the images are resized to $256 \times 256$, then are randomly cropped to $224 \times 224$ and the data is normalized using the mean and standard deviation statistics of the train inputs; no additional augmentation is done. At inference time, the images are first scaled to $256 \times 256$, center cropped to $224 \times 224$ and then normalized.

We evaluate the performance of our training method on two architectures AlexNet and Resnet-18 (He et al., 2016). Following previous work, we used batch-normalization before each activation function. Additionally, we keep the first and last layers to be in full precision, as we lose 2-3% accuracy otherwise. This approach is followed by other binary methods that we compare to (Hubara et al., 2016; Rastegari et al., 2016; Tang et al., 2017). The results are summarized in Table 2. In all the experiments involving $R_1$ regularization we set the $\lambda$ to $10^{-7}$ and $R_2$ regularization to $10^{-6}$. Also, in every network, the scales are introduced per filter in convolutional layers, and per column in fully connected layers. The weights are initialized using a pre-trained model with $\text{htan}$ activation function as done in Liu et al. (2018). Then the learning rate for AlexNet is set to $2.33 \times 10^{-3}$ and multiplied by 0.1 at the $12^{th}$, $18^{th}$ epoch for a total of 25 epochs trained. For the 18-layer ResNet the learning rate is started from 0.01 and multiplied by 0.1 at $10^{th}, 20^{th}, 30^{th}$ epoch.

## 4.3 DISCUSSION

We proposed two regularization terms (4) and (5) and an activation term (2) with a trainable parameter $\beta$. We run several experiments to better understand the effect of the different modifications to the training method, especially using different regularization and asymptote parameters $\beta$. The parameter $\beta$ is trainable and would add one equation through back-propagation. However, we fixed $\beta$ throughout our experiments to explicit values. The results are summarized in Table 2.

Through our experiments, we found that adding regularizing term with heavy penalization degrades the networks ability to converge, as the term would result in total loss be largely due to the regu-

Table 2: Top-1 and top-5 accuracies (in percentage) on ImageNet dataset, of different combinations of the proposed technical novelties on different architectures.

| Regularization | Activation | AlexNet | | Resnet-18 | |
|---|---|---|---|---|---|
| $R_1$ | $SS_5$ | 46.11 | 75.70 | - | - |
| | $SS_{10}$ | 46.08 | 75.75 | 51.13 | 74.94 |
| | htanh | 41.58 | 69.90 | 50.72 | 73.48 |
| $R_2$ | $SS_5$ | 45.62 | 70.13 | 53.01 | 72.55 |
| | $SS_{10}$ | 45.79 | 75.06 | 49.06 | 70.25 |
| | htanh | - | - | 48.13 | 72.72 |
| None | $SS_5$ | 45.25 | 75.30 | 43.23 | 68.51 |
| | $SS_{10}$ | 45.60 | 75.30 | 44.50 | 64.54 |
| | $SS_{20}$ | 44.03 | 68.30 | 44.74 | 69.62 |
| | htanh | 39.18 | 69.88 | 42.46 | 67.56 |

larizing term and not the target cross entropy loss. Similarly, the regularizing term was set to small values in Tang et al. (2017). As a result, we set $\lambda$ with a reasonably small value $10^{-5} - 10^{-7}$, so that the scales move slowly as the weights gradually converge to stable values. Some preliminary experimentation was to gradually increase the regularization with respect to batch iterations updates done in training, though this approach requires careful tuning and was not pursued further.

From Table 2, and referring to networks without regularization, we see the benefit of using Swish-Sign approximation versus the STE. This was also noted in Liu et al. (2018), where their second approximation provided better results. There is not much difference between using $R_1$ versus $R_2$ towards model generalization although since the loss metric used was the cross-entropy loss, the order of $R_1$ better matches the loss metric. Lastly, it seems moderate values of $\beta$ is better than small or large values. Intuitively, this happens because for small values of $\beta$, the gradient approximation is not good enough and as $\beta$ increases the gradients become too large, hence small noise could cause large fluctuations in the sign of the weights.

We did not compare our network with that of Liu et al. (2018) as they introduce a shortcut connection that proves to help even the full precision network. As a final remark, we note that the learning rate is of great importance and properly tuning this is required to achieve convergence. Table 3 summarizes the best results of the ablation study and compares with BinaryNet, XNOR-Net, and ABC-Net.

Table 3: Comparison of top-1 and top-5 accuracies of our method BNN+ with BinaryNet, XNOR-Net and ABC-Net on ImageNet, summarized from Table 2. The results of BNN, XNOR, & ABC-Net are reported from the corresponding papers (Rastegari et al., 2016; Hubara et al., 2016; Tang et al., 2017). Results for ABC-NET on AlexNet were not available, and so is not reported.

| | | BNN+ | BinaryNet | XNOR-Net | ABC-Net | Full-Precision |
|---|---|---|---|---|---|---|
| AlexNet | Top-1 | **46.1**% | 41.2% | 44.2% | - | 56.6% |
| | Top-5 | **75.7**% | 65.6% | 69.2% | - | 80.2% |
| Resnet-18 | Top-1 | **53.0**% | 42.2% | 51.2% | 42.7% | 69.3% |
| | Top-5 | 72.6% | 67.1% | **73.2**% | 67.6% | 89.2% |

## 5 CONCLUSION

To summarize we propose three incremental ideas that help binary training: i) adding a regularizer to the objective function of the network, ii) trainable scale factors that are embedded in the regularizing term and iii) an improved approximation to the derivative of the sign activation function. We obtain competitive results by training AlexNet and Resnet-18 on the ImageNet dataset. For future work, we plan on extending these to efficient models such as CondenseNet (Huang et al., 2018), MobileNets (Howard et al., 2017), MnasNet (Tan et al., 2018) and on object recognition tasks.

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
