# OpenReview forum: "BNN+: Improved Binary Network Training"
_ICLR.cc/2019/Conference_

### Official Review · AnonReviewer1 · 2018-10-30
**Borderline paper -- OK empirical results but weak in most other regards**

**Rating:** 4
**Confidence:** 4

**Review:**

Summary:
This paper presents three small improvements for training binarized neural networks: (1) a modified straight-through estimator, (2) a novel regularizer to push weights to +/- 1, and (3) the use of scaling factors for the binarized weights. Using the methods presented, the validation accuracies on ImageNet and CIFAR-10 are improved by just under 2 percentage points.

  Pros:
    - Decent improvement in the performance of the binarized network in the end
    - The presented regularizers make sense and seem effective. The modified straight-through estimator seems reasonable as well, although the authors do not compare to recent work with a similar adjustment.

  Cons:
    - The paper is poorly written and confusing. It reads as if it was written in one pass with no editing or re-writing to clarify contributions or key points, or ensure consistency.
    - While the final numbers are acceptable, the experiments themselves could be stronger and could be presented more effectively.
   - The novelty of the scale factors is questionable.


Questions and comments:

1. How exactly is the SS_\beta activation used? It is entirely unclear from the paper, which contradicts itself in multiple ways. Is SS_\beta used in the forward pass at all for either the weight or activation binarization? Or is only its derivative used in the backward pass? If the latter, then you are not replacing the activation anywhere but are simply using a different straight-through estimator in place of the saturated straight-through estimator (e.g., see [1]).
   (a) At the beginning of Section 3.3, you say that you modify the training procedure by replacing the sign binarization with the SS_\beta activation. This sounds like it is referring to the activation function at each layer; however, the pseudocode says that you are using sign() as the per-layer activation.
   (b) Further, Figure 4 shows that you are using the SS_\beta function to do weight binarization. However, again, the pseudocode shows that you are using sign() to do the weight binarization.

2. In [1], the authors used a similar type of straight-through estimator (essentially, the gradient of tanh instead of hard_tanh) and found that to be quite effective. You should compare to their method. Also, it's possible that SS_\beta reduces to tanh for some choice of \beta -- is this true?

3. The use of scale factors seems to greatly increase the number of parameters in the network and thus greatly decrease the compression benefits gained by using binarization, i.e., you require essentially #scale_factors =  a constant factor times the number of actual parameters in the network (since you have a scale factor for each convolutional filter and for each column of each fully-connected layer). As a result of this, what is the actual compression multiplier that your network achieves relative to the original network?

4. For the scale factor, how does yours differ from that used in Rastegari et al. (2016)? It seems the same but you claim that it is a novel contribution of your work. Please clarify.

5. Why did learning \beta not work? What was the behavior? What values of \beta did learning settle on?

6. I assume that for each layer output y_i = f(W_i x_i), the regularizer is applied as R(y_i) while at the same time y_i is passed to the next layer -- is this correct? The figures do not clearly show this and should be changed to more clearly show how the regularizer is computed and used, particularly in relation to the activation.

7. In the pseudocode:
   (a) What does "mostly bitwise operations" mean? Are some floating point?
   (b) Is this the shift-based batchnorm of Hubara et al. (2016)?

8. For Table 1:
   (a) I assume these are accuracies? The caption should say.
   (b) Why are there no comparisons to the performance of other methods on this dataset?
   (c) Any thoughts as to why your method performs better than the full-precision method on this dataset for VGG?

8. For Table 2:
   (a) Does Table 2 show accuracies on ImageNet? You need to make this clear in the caption.
   (b) What type of behavior do the runs that do not converge show? This seems like a learning rate problem that is easily fixable. Are there no hyperparameter values that allow it to converge?
   (c) What behavior do you see when you use SS_1 or SS_2, i.e., \beta = 1 or \beta = 2? Since lower \beta values seem better.
   (d) The regularization seems to be the most useful contribution -- do you agree?
   (e) Why did you not do any ablations for the scale factor? Please include these as well.

9. For Table 3, did you compute the numbers for the other approaches or did you use the numbers from their papers? Each approach has its own pros and cons. Please be clear.

10. Are there any plots of validation accuracy versus epoch/time for the different algorithms in order to ensure that the reported numbers were not simply cherry-picked from the run? I assume that you simply used the weights from the end of the 50th epoch -- correct?

11. Is there evidence for your introductory claims that 'quantizing weights ... make neural networks harder to train due to a large number of sign fluctuations' and 'maintaining a global structure to minimize a common cost function is important' ? If so, you should cite this evidence. If not, you should make it clear that these are hypotheses.

12. Why are there not more details about the particular architectures used? These should be included in the appendices to aid those who would like to rerun your experiments. In general, please include more experiment details in the body or appendices.


Detailed comments:
- R(l) is not defined in Figure 1 and thus is confusing. Also, its replacement of 'Error' from the original figure source makes the figure much more confusing and less clear.

- Typos:
   - 'accustomed' (p.1)
   - 'the speed by quantizing the activation layers' doesn't make sense (p.1)
   - 'obtaining' (p.4)
   - 'asymmetric' doesn't make sense because these are actually symmetric functions across the y-axis (p.4)
   - 'primary difference is that this regularization ...' --> 'primary difference is that their regularization ...' (p.4)
   - 'the scales with 75th percentile of the absolute value ... ' is very confusing and unclear (p.7)
   - 'the loss metric used was the cross-entropy loss, the order of R_1.' I do not know what you're trying to say here (p.8)

- Citations: Fix the capitalization issues, typos, and formatting inconsistencies.


[1]  Friesen and Domingos. Deep Learning as a Mixed Convex-Combinatorial Optimization Problem. ICLR 2018.


-------------------

After reading the author response, I do not think the paper does a sufficient job of evaluating the contributions or comparing to existing work. The authors should run ablation experiments, compare to existing work such as [1], and evaluate on additional datasets. These were easy tasks that could have been done during the review period but were not.

If I wanted to build on top of this paper to train higher accuracy binary networks, I would have to perform all of these tasks myself to determine which contributions to employ and which are unnecessary. As such, the paper is currently not ready for publication.

---

> ### Author Response · Authors · 2018-11-20
> **To Reviwer #3 (4/4)**
>
>
> *comment: “Are there any plots of validation accuracy versus epoch/time for the different algorithms in order to ensure that the reported numbers were not simply cherry-picked from the run? I assume that you simply used the weights from the end of the 50th epoch -- correct?”
>
>
> Yes we simply evaluated the model at the end of the training procedure.
>
> ----------------------------------------------------------------------------------------------------------------
> *comment: “Is there evidence for your introductory claims that 'quantizing weights ... make neural networks harder to train due to a large number of sign fluctuations' and 'maintaining a global structure to minimize a common cost function is important' ? If so, you should cite this evidence. If not, you should make it clear that these are hypotheses.”
>
> This intuition and reasoning was provided in [3], thanks for pointing this out we made sure to cite the paper.
>
> Concerning “maintaining a global structure to minimize a common cost function” we re-wrote it as:
>
> “How to quantize the weights locally, and maintaining a global structure to minimize common cost function is important[4]”
>
>
> [3] Lin, Xiaofan, Cong Zhao, and Wei Pan. "Towards accurate binary convolutional neural network." Advances in Neural Information Processing Systems. 2017.
> [4]Li, Hao, et al. "Training quantized nets: A deeper understanding." Advances in Neural Information Processing Systems. 2017.
>
> -------------------------------------------------------------------------------------------------------------------------------------------------------------------
> *comment: “Why are there not more details about the particular architectures used? These should be included in the appendices to aid those who would like to rerun your experiments. In general, please include more experiment details in the body or appendices.”
>
> We added more explanations on the architectures and experiments in the corresponding sections.
>
> We train both, AlexNet \citep{krizhevsky2012imagenet}, and VGG \citep{Simonyan2014VeryDC} using the ADAM \citep{Kingma2014AdamAM} optimizer. The architecture used for VGG is conv(256)-conv(256-conv(512-conv(512-conv(1024)-conv(1024-fc(1024-fc(1024) where conv(\cdot) is a convolutional layer, and fc(\cdot) is a fully connected layer. The standard 3\times3 filters are used in each layer. We also add a batch normalization layer (Ioffe2015BatchNA) prior to activation. For AlexNet, the architecture from (Krizhevsky2014OneWT) is used, and batch normalization layers are added prior to activations. We use a batch size of 256 for training. Many learning rates were experimented with such as 0.1, 0.03, 0.001,  etc, and the initial learning rate for AlexNet was set to 10^{-3}, and 3 \times 10^{-3} for VGG.
>
> - Typos:
> Thank you for pointing these out. We made sure to correct all typos and unclear sentences in the revised version of the paper.

---

> ### Author Response · Authors · 2018-11-20
> **To Reviewer #3 (3/4)**
>
> *comment: For Table 1:
>    (a) "I assume these are accuracies? The caption should say."
>
> We modified the caption as follows:
> “Accuracy results on test set for CIFAR10 dataset, using Manhattan regularization function (\ref{l1reg}) with AlexNet and VGG.”
>
>    (b) "Why are there no comparisons to the performance of other methods on this dataset?"
>
> Cifar10 consists of low resolution images and limited number of classes hence it is not a challenging dataset. Accordingly in order to show the empirical results for our comparisons, we decided to forgo comparison on this task and instead focus on ImageNet.
> We thought comparison on CIFAR10 does not provide fair comparison as it is not a challenging dataset. Instead it makes sense to do so on a harder dataset such as Imagenet.
>
>    (c) "Any thoughts as to why your method performs better than the full-precision method on this dataset for VGG?"
>
> VGG network is over-parameterized and given the simplicity of CIFAR10 dataset, binarization helps regularize the network and avoid overfitting.
>
> --------------------------------------------------------------------------------------------------------------
> *comment: For Table 2:
>    (a) "Does Table 2 show accuracies on ImageNet? You need to make this clear in the caption."
>
> Table 2 we modified the caption as follows:
>
> “ImageNet Top-1 and top-5 accuracies (in percentage) of different combinations of the proposed technical novelties on different architectures. Some architectures were harder to train and did not converge within the time frame of the others, and so is not reported.”
>
>    (b) What type of behavior do the runs that do not converge show? This seems like a learning rate problem that is easily fixable. Are there no hyperparameter values that allow it to converge?
>
> Table 2 is a computationally time consuming task, as we are training the networks on the ImageNet dataset. As a result, for those two specific table values our experiments were not run. This sentence hadn’t been worded appropriately. Also we would like to point that there are no convergence issues with the method.
>
>    (c) "What behavior do you see when you use SS_1 or SS_2, i.e., \beta = 1 or \beta = 2? Since lower \beta values seem better."
>
> Lower \beta value do not necessarily mean better results. We modified the sentence in the text
>
> “Lastly, it seems smaller (moderate) values of β is better than larger values.”
>
> In theory, higher \beta value mean better approximation of the derivative of the sign function. However, there is a trade-off between approximation of the derivative of the sign function and the gradient value. So, moderate values of \beta seem better. So, we presented SS_5 and SS_10.
>
>    (d) "The regularization seems to be the most useful contribution -- do you agree?"
>
> Our contributions are incremental, like just about any contributions. E.g., ReLUs were a small modification of activation functions, yet ReLUs had a huge impact. The regularization functions introduced in this paper are two out of many functions we can use for quantizing BNN. Including the scaling factors in the regularization function. As a result, learning the scales through back-propagation is a contribution and useful since it gives a flexibility to the method to adapt to the data. Finally, based on our experimental results summarized in Table 2, our approximation of the derivative of the sign function proves to be useful empirically l since it gives better results than the straight through estimator.
>
>
>    (e) "Why did you not do any ablations for the scale factor? Please include these as well."
>
> The scaling factors are parameters estimated along with the weights. By our comparison with the BNN method we demonstrated the efficacy of using the suggested scales.
>
> --------------------------------------------------------------------------------------------------------------
> *comment: For Table 3, did you compute the numbers for the other approaches or did you use the numbers from their papers? Each approach has its own pros and cons. Please be clear.
>
> We have a re-implementation of BNN, and for XNOR Net we decided to cite the accuracies in the paper instead. We have added the following to the caption of the paper:
>
> “Comparison of top-1 and top-5 accuracies of our method BNN$+$ with BinaryNet, XNOR-Net and ABC-Net on ImageNet, summarized from Table \ref{tab:ablation}. Results are cited from the corresponding papers.”

---

> > ### Comment · AnonReviewer1 · 2018-11-25
> > **Reviewer response**
> >
> > "We thought comparison on CIFAR10 does not provide fair comparison as it is not a challenging dataset. Instead it makes sense to do so on a harder dataset such as Imagenet."
> >
> > While CIFAR10 is easier than ImageNet, performance on it has not yet necessarily saturated for binarized networks, and using a smaller dataset allows one to run experiments much more quickly. This can be extremely useful for running things like ablation experiments, which are crucial when multiple changes are introduced in a single paper. Further, it is always useful to evaluate on multiple datasets, to ensure that one hasn't simply overfit to a single dataset.
> >
> > --------
> >
> > "Lower \beta value do not necessarily mean better results. "
> >
> > Does this mean that you ran experiments with these values of \beta? The statement you made is extremely vague.
> >
> > --------
> >
> > "The scaling factors are parameters estimated along with the weights. By our comparison with the BNN method we demonstrated the efficacy of using the suggested scales."
> >
> > How can this be a demonstration of the efficacy of using the suggested scales when the other contributions were also included in this test (i.e., the regularization and the swish-as-STE approximation). It is thus not clear what the benefit of any of these is in isolation, which is the entire point of running ablation experiments.

---

> ### Author Response · Authors · 2018-11-20
> **To Reviewer #3 (2/4)**
>
> *comment: For the scale factor, how does yours differ from that used in Rastegari et al. (2016)? It seems the same but you claim that it is a novel contribution of your work. Please clarify.
>
> In Rastegari et al. (2016) the parameters are estimated dynamically given the weights of the network. Hence in training, on each pass they are updated accordingly. Whereas in our work we introduce the scaling factors in the regularization function.
>
> This follows as the scaling factors introduced by Rastegari et al. (2016) are estimated in a 2-stage fashion. First they find the weights and second, they solve an optimization problem (L2 norm of the difference between full-precision weights and scaling factor times binary weights) in order to estimate the scaling factors. So, the estimated scaling factor is the mean of absolute values of the weights.
>
> In our work, there is a difference in how the scales are formulated. We introduce the scales into a regularization function constructed specifically for a BNN. This class of regularization functions can be written to
>
> R(w) = |scaling_factor - |weights| |^p
>
> where p=1 and 2 in the paper. Instead of having two separate optimization problems, we back-propagate the updates to the scales in order to minimize the loss function plus the regularization term.
>
> Depending on the regularization term used, the scaling factors estimation falls into either mean of absolute values of the weights (p=2) or median of absolute values of the weights (p=1). As a result, this could also be seen as a generalization of Rastegari et al. (2016)’s scaling factor.
>
> ----------------------------------------------------------------------------------------------------------------
> *comment: Why did learning \beta not work? What was the behavior? What values of \beta did learning settle on?
>
>
> Learning beta adds only one equation to back-propagation in our experiments we fixed  beta to have explicit value. We did not extensively try to experiment with learning beta. We changed the following sentence accordingly:
>
> “The parameter $\beta$ could be learn-able , and would add one equation only to back-propagation. However,  we fixed $\beta$ through out our experiments. The results are summarized in Table \ref{tab:ablation}”
>
> We did few experiments having beta trainable though it was not conclusive. Accordingly, we decided to leave it for future work. For instance one can investigate changing the beta parameter dynamically as training progresses, perhaps by starting with a smaller beta and moving towards larger beta.
>
> -------------------------------------------------------------------------------------------------------------
> *comment: I assume that for each layer output y_i = f(W_i x_i), the regularizer is applied as R(y_i) while at the same time y_i is passed to the next layer -- is this correct? The figures do not clearly show this and should be changed to more clearly show how the regularizer is computed and used, particularly in relation to the activation.
>
> Referring to equation (6) of section 3.3, the regularizer is applied as a function of the weights and scaling factor (R(W_l, \alpha_l)), which is then added to the loss function. This total loss is used to optimize the network.
>
> From section 3.3:
> “J(W, b) = L(W, b) + \lambda \sum_{l} R(W_l,\alpha_l)
>
> where L(W, b) is the cost function, W and b are the sets of all weights and biases in the network, W_l is the set weights at layer l and \alpha_l is the corresponding scaling factor. Here, R(.) is the regularization function 4 or 5.”
>
> --------------------------------------------------------------------------------------------------------------
> *comment: In the pseudocode:
>    (a) What does "mostly bitwise operations" mean? Are some floating point?
> Here by mostly bitwise operation in algorithm 1 we mean that, the only floating operation is the multiplication of the scales, into the W^bx^b (line 6 in forward pass of algorithm 1)
>
>    (b) Is this the shift-based batchnorm of Hubara et al. (2016)?
>     No. But, shift-based batchnorm of Hubara et al. (2016) is orthogonal to our methodology. Shift-based batchnorm is only useful if you want to speed up the training. At run-time, you can fold the vanilla batchnorm operations into a simple threshold function.

---

> ### Author Response · Authors · 2018-11-20
> **To Reviewer #3 (1/4)**
>
> Thank you for your constructive feedback. Please find below a point to point response.
>
> -----------------------------------------------------------------------------------------------------------------
> *comment: How exactly is the SS_\beta activation used? It is entirely unclear from the paper, which contradicts itself in multiple ways. Is SS_\beta used in the forward pass at all for either the weight or activation binarization? Or is only its derivative used in the backward pass? If the latter, then you are not replacing the activation anywhere but are simply using a different straight-through estimator in place of the saturated straight-through estimator (e.g., see [1]).
>
> In the forward pass the sign function is used, and in the backward pass we use the derivative of SwishSign function as the approximation in the backward pass. We corrected the sentence in the text:
>
> “Combining both the regularization and activation ideas, we modify the training procedure by replacing the \sign backward approximation binarization with that of the derivative of the SS_\beta activation (2).”
>
>      b) Due to the confusion introduced by this figure, we removed it accordingly.
>
> -----------------------------------------------------------------------------------------------------------------
> *comment: In [1], the authors used a similar type of straight-through estimator (essentially, the gradient of tanh instead of hard_tanh) and found that to be quite effective. You should compare to their method. Also, it's possible that SS_\beta reduces to tanh for some choice of \beta -- is this true?
>
> [1] is concerned with solving a combinatorial optimization problem for hard thresholding activation units. In their work they keep the weights to full precision values, and only limit the activations of units to binary values. Our primary motivation behind SS_beta was to define a class of functions for which the derivatives are different approximations of the sign function. In [1], the approximation is fixed and therefore less flexible than ours. In the ablation study, we see that for different values of beta, the accuracy changes.The SS_\beta does not reduce to the tanh function. The tanh function is similar to that of the sigmoid, although in the case of SS_1 they look similar, in the swish function there is a subtle difference in that there is a bump at -2.4/beta and +2.4/beta which helps with learning (gradient flow, saturation at later point). Further, one of the major difference between signswish and tanh is that signswish is non monotonous.
>
> -------------------------------------------------------------------------------------------------------------
> *comment: The use of scale factors seems to greatly increase the number of parameters in the network and thus greatly decrease the compression benefits gained by using binarization, i.e., you require essentially #scale_factors =  a constant factor times the number of actual parameters in the network (since you have a scale factor for each convolutional filter and for each column of each fully-connected layer). As a result of this, what is the actual compression multiplier that your network achieves relative to the original network?
>
> The use of scale factors does reduce the actual compression, but not significantly. For example in the case of AlexNet,
>
> conv 64 - conv 192 - conv 384 - conv 256 - conv 256 - FC 4096 - FC 4096
>
> The number of parameters in the original network are ~ 6M/32 = 187500
> The number of scales introduced are (192+384+256+256+4096+4096) = 9280
> The compression with the addition of scales becomes ~ 31.5
>
> The additional overhead of the scales is less than ~2 %.
>
> At inference time, one can fold the batch norms onto the scaling factors, thus removing the batchnorms operations and their parameters. This has the similar effect as in [2]. As a result although we introduce scaling factors, we remove the batch norm parameters and division operation.
>
> [2] Rastegari, Mohammad, et al. "Xnor-net: Imagenet classification using binary convolutional neural networks." European Conference on Computer Vision. Springer, Cham, 2016.

---

> > ### Comment · AnonReviewer1 · 2018-11-25
> > **Reviewer response**
> >
> > "[1] is concerned with solving a combinatorial optimization problem for hard thresholding activation units. ..."
> >
> > No, [1] is concerned with the same problem as this paper -- training neural networks with binarized activations -- they simply formulate the problem as a mixed convex-combinatorial optimization problem to better understand and justify their approach and the STE. While they do not use binarized weights, other papers (see third paragraph of intro of [3] and citations within) have shown that the activation binarization causes far more accuracy loss and is thus the critical component of binarization. Further, activation binarization is orthogonal to weight binarization so it would be trivial to test their activation as well.
> >
> > The issue here is that you cannot simply claim that your approach is better, even if it is more flexible (and some would argue that introducing additional hyperparameters makes training more challenging), unless you compare directly to existing work. Your experiments loosely show that lower values of \beta outperform higher values of \beta. You say this is not the case but provide no clear evidence for the reader. Further, you say that the bump in the swish helps with learning but, again, this is not clearly demonstrated in the experiments. Using \beta=1 would give an STE with slope=1 at 0 (which tanh has, as used in [1]) but with the non-monotonicity that tanh does not have. This would provide a direct comparison between their approach and yours (although, even a direct comparison using \beta=5 would be fine too).
> >
> > You have introduced multiple changes to existing architectures but not clearly evaluated them to show their respective contributions and utility. You do not compare to existing work ([1], etc.) that makes a very similar change to the approximation of the derivative of the activation.
> >
> > [3] Deep Learning with Low Precision by Half-wave Gaussian Quantization. Cai, He, Sun, and Vasconcelos (2017).

---

> ### Author Response · Authors · 2018-11-26
> **Authors Response to Reviewer #3**
>
>
> We are saddened to see that our response convinced you to reduce your score from 5 to 4.
>
> We agree that [1] proposes an interesting method, which is somewhat related to ours.
>
> However, we also believe that it would be very long and difficult for us to compare this method with ours:
>
> (A) [1] binarizes the activations, but not the weights.
> Although we agree with you that binarizing activations is harder than binarizing weights, we believe that binarizing both is harder than binarizing only the activations.
> There is no guarantee that [1] would work with binary weights.
>
> (B) [1] does not have ImageNet results.
> Although CIFAR-10 is certainly an interesting benchmark,
> we believe that ImageNet is more challenging.
> There is no guarantee that [1] would work on ImageNet.
>
> We disagree with your assertion that we did not clearly evaluate the respective contributions of the 2 changes we introduced.
> Our table 2 shows that our improved "swish" straight-through estimator (STE) performs better than the "hardtanh" STE, with or without our new regularization term.
> Our table 2 also shows that our new regularization term improves performance
> independently of whether the "swish" or the "hardtanh" STE is used.

---

> > ### Comment · AnonReviewer1 · 2018-11-26
> > **Reviewer's response**
> >
> > I am now skeptical that you even read [1]. It would be easy to implement and compare to their approach, and they do report ImageNet results. Here is a link if you need: https://openreview.net/forum?id=B1Lc-Gb0Z. I believe that they also provide code to run their method. In more detail:
> >
> > (A) Absolutely, there is no guarantee that [1] would work with binary weights but also no reason to think that it wouldn't for the reasons outlined previously. It would be trivial for you to run one more evaluation using your approach with their STE (essentially, just a tanh) to determine if there is actually any merit to your STE. Alternatively, it would be trivial for you to run one more evaluation using your STE with their approach, for the same effect.
> >
> > (B) This is clearly false. [1] has ImageNet results. Half of Section 4 in [1] is dedicated to ImageNet results.
> >
> > To reiterate, my point is that [1] also improves on the hardtanh STE. Thus, while your method may also improve on the hardtanh STE, it does not necessarily improve on the method of [1], which already improved on the hardtanh STE. Since your STE has a similar softening as that of [1], it is not clear whether your introduced STE is achieving the same thing that [1] achieved or if there is merit to your claim that the non-monotonicity is important. While, absolutely, this claim could be true, you have not demonstrated that it is; instead, you have simply claimed that it is while comparing only to older work.

---

> > > ### Author Response · Authors · 2018-11-26
> > > **Author's Response**
> > >
> > > Sorry if there was a misunderstanding about (B).
> > >
> > > Here, we meant that it is unclear whether the method proposed in [1] is useful for ImageNet with 1-bit activations. In their table 1, on ResNet-18 with 1-bit activations (called Sign in the table), the proposed method FTP-SH actually performs worse than the htanh STE (SSTE in the table).
> > >
> > > Our opinion is that the method introduced in [1] really shines with 2 or 3-bit activations, and full precision weights. As a result, it may be difficult to do a fair comparison with our work.
> > > We nonetheless believe that the article is a great addition to our related work section.
> > >
> > > Further, although we did not re-implement the exact recursive mini-batch algorithm of [1], we already did run experiments using a smooth tanh STE on CIFAR-10 (with Alexnet) as well as our SS1, SS2 and SS5. The tanh STE results were inferior to that of the SS5 STE by a ~2% margin. We plan to add those results to our table 1.
> > >
> > > These experiments suggest that the non-monotonicity and the flexibility of our new STE are beneficial compared to using a simpler smooth STE function.

---

### Official Review · AnonReviewer2 · 2018-11-02
**Impressive results for binarized neural networks by combining existing ideas**

**Rating:** 6
**Confidence:** 3

**Review:**

The authors of this paper aim to reduce the constraints required by neural networks so they can be evaluated on lower-power devices. Their approach is to quantize weights, i.e. rounding weights and hidden units so they can be evaluated using bit operations. There are many challenges in this approach, namely that one cannot back-propagate through discrete weights or discrete sign functions. The authors introduce an approximation of the sign function, which they call the SignSwish, and they back-propagate through this, quantizing the weights during the forward pass. Further, they introduce a regularization term to encourage weights to be around learned scales. They evaluate on CIFAR-10 and Imagenet, surpassing most other quantization methods.

The paper is pretty clear throughout. The authors do a good job of motivating the problem and placing their approach in the context of previous work. I found Figures 1 and 2 helpful for understanding previous work and the SignSwish activation function, respectively. However, I did not get much out of Figures 3 or 4. I thought Figure 3 was unnecessary (it shows the difference between l1 and l2 regularization), and I thought the psuedo-code in Algorithm 1 was a lot clearer than Figure 4 for showing the scaling factors. Algorithm 1 helped with the clarity of the approach, although it left me with a question: In section 3.3, the authors say that they train by "replacing the sign binarization with the SS_\beta activation" and that they can back-propagate through it. However, in the psuedo-code it seems like they indeed use the sign-function in the forward-pass, replacing it with the signswish in the backward pass. Which is it?

The original aspects of their approach are in introducing a new continuous approximation to the sign function and introducing learnable scales for l1 and l2 regularization. The new activation function, the SignSwish, is based off the Swish-activation from Ramachandran et al. (2018). They modify it by centering it and taking the derivative. I'm not sure I understand the intuition behind using the derivative of the Swish as the new activation. It's also unclear how much of BNN+'s success is due to the modification of the Swish function over using the original Swish activation. For this reason I would've liked to see results with just fitting the Swish. In terms of their regularization, they point out that their L2 regularization term is a generalization of the one introduced in Tang et al. (2017). The authors parameterize the regularization term by a scale that is similar to one introduced by Rastegari et al. (2016). As far as I can tell, these are the main novel contributions of the authors' approach.

This paper's main selling point isn't originality -- rather, it's that their combination of tweaks lead to state-of-the-art results. Their methods come very close to AlexNet and VGG in terms of top-1 and top-5 CIFAR10 accuracy (with the BNN+ VGG even eclipsing the full-precision VGG top-1 accuracy). When applied to ImageNet, BNN+ outperforms most of the other methods by a good margin, although there is still a lot of room between the BNN+ and full-precision accuracies. The fact that some of the architectures did not converge is a bit concerning. It's an important detail if a training method is unstable, so I would've liked to see more discussion of this instability. The authors don't compare their method to the Bi-Real Net from Liu et al. (2018) since it introduces a shortcut connection to the architecture, although the Bi-Real net is SOTA for Resnet-18 on Imagenet. Did you try implementing the shortcut connection in your architecture?

Some more minor points:
- The bolding on Table 2 is misleading. It makes it seem like BNN+ has the best top-5 accuracy for Resnet-18, although XNOR-net is in fact superior.
- It's unclear to me why the zeros of the derivative of sign swish being at +/- 2.4beta means that when beta is larger, we get a closer approximation to the sign function. The derivative of the sign function is zero almost everywhere, so what's the connection?
- Is the initialization of alpha a nice trick, or is it necessary for stable optimization? Experiments on the importance of alpha initialization would've been nice.

PROS:
- Results. The top-1 and top-5 accuracies for CIFAR10 and Imagenet are SOTA for binarized neural networks.
- Importance of problem. Reducing the size of neural networks is an important direction of research in terms of machine learning applications. There is still a lot to be explored.
- Clarity: The paper is generally clear throughout.

CONS:
-Originality. The contributions are an activation function that's a modification of the swish activation, along with parameterized l1 and l2 regularization.
-Explanation. The authors don't provide much intuition for why the new activation function is superior to the swish (even including the swish in Figure 2 could improve this). Moreover, they mention that training is unstable without explaining more.

---

> ### Author Response · Authors · 2018-11-20
> **To Reviewer #2 (3/3)**
>
> *comment: "The authors don't compare their method to the Bi-Real Net from Liu et al. (2018) since it introduces a shortcut connection to the architecture, although the Bi-Real net is SOTA for Resnet-18 on Imagenet. Did you try implementing the shortcut connection in your architecture?"
>
>
> The connection was not added, as suggested in bi-real net. Here our main objective was to improve the training mechanism for Binary Networks. As future work we can investigate more efficient architectures for binarized neural networks, such as condensenet as opposed to residual networks due to their summation operator rids of too much information, whereas in condensenet the activations are appended hence information is maintained  across layers.

---

> ### Author Response · Authors · 2018-11-20
> **To Reviewer #2 (2/3)**
>
> *comment: “It's unclear to me why the zeros of the derivative of sign swish being at +/- 2.4beta means that when beta is larger, we get a closer approximation to the sign function. The derivative of the sign function is zero almost everywhere, so what's the connection?”
>
> This is a property of the swish sign function, which locates the max and min at +/- 2.4/beta. We thought it’d be interesting to point this out, and by this we meant that by increasing beta, one can adjust the locations at which the gradients start saturating. Also, it is easy to show that when beta tends to infinity, sign swish converges to the sign function. We added an explanation in the text:
>
> “Note that the derivative d/dxSS_\beta(x) is zero at two points, controlled by \beta. Indeed, it is simple to show that the derivative is zero for x \approx \pm 2.4 / \beta. By adjusting this parameter beta, it is possible to adjust the location at which the gradients start saturating. In contrast to the STE estimators, where it is fixed. Thus, the larger \beta is, the closer the approximation is to the derivative of the \sign function”
>
> ----------------------------------------------------------------------------------------------------------------
> *comment: I'm not sure I understand the intuition behind using the derivative of the Swish as the new activation. It's also unclear how much of BNN+'s success is due to the modification of the Swish function over using the original Swish activation. For this reason I would've liked to see results with just fitting the Swish.
>
> As explained in our answer to the previous comment, the swish is not bounded on the right side.
>
> “where \sigma(z) is the sigmoid function and the scale \beta > 0 controls how fast the activation function asymptotes to -1 and +1. The \beta parameter can be learned by the network or be hand-tuned as a hyperparameter. As opposed to the Swish function, where it is unbounded on the right side, the modification make it bounded and a valid approximator of the \sign function. As a result, we call this activation SignSwish, and its gradient is”
> ----------------------------------------------------------------------------------------------------------------
>
> *comment: In terms of their regularization, they point out that their L2 regularization term is a generalization of the one introduced in Tang et al. (2017). The authors parameterize the regularization term by a scale that is similar to one introduced by Rastegari et al. (2016). As far as I can tell, these are the main novel contributions of the authors' approach
>
> The notion of regularization as well as that of scaling factors are not new in the literature of binary neural networks (BNN). But, the regularization introduced in Tang et al. (2017) is different from ours because it does not penalize the weights outside of [-1;1], it is a quadratic function (1 - w^2) and does not include the notion of scaling factor. For the scaling factors introduced by Rastegari et al. (2016), they estimate them in a 2-stage fashion. They first find the weights and second, they solve an optimization function (L2 norm of the difference between full-precision weights and scaling factor times binary weights) in order to estimate the scaling factors. This results in the scaling factor is the mean of absolute values of the weights.
>
> In our work, we introduce a new approach to the quantization of BNNs. Our novelty is the introduction of the scaling factor into a regularization function constructed for a BNN (class of regularization functions R(w) = |scaling_factor - |weights| |^p where p=1 and 2 in the paper) , as well as an adaptive approximation (using a parameter) of the derivative of the sign function. Instead of having two separate optimization problems, we only need back-propagation in order to minimize the loss function plus the regularization term in order to estimate the binary weights as well as the scaling factors. Depending on the regularization term used, the scaling factors estimation falls into either mean of absolute values of the weights (p=2) or median of absolute values of the weights (p=1).
>
> ----------------------------------------------------------------------------------------------------------------
> *comment: The authors don't provide much intuition for why the new activation function is superior to the swish (even including the swish in Figure 2 could improve this). Moreover, they mention that training is unstable without explaining more.
>
> I believe we addressed this in prior comments. There is no new activation function but a new approximation of the sign function derivative. Hence, the swish function is not appropriate, but our modification is. We point out the relationship with the swish function is due to our SignSwish function being modification to the derivative of the swish function.

---

> ### Author Response · Authors · 2018-11-20
> **To Reviewer #2 (1/3)**
>
> Thank you for your constructive feedback. Please find below a response to the comments.
> -----------------------------------------------------------------------------------------------------------------
> *comment: However, I did not get much out of Figures 3 or 4. I thought Figure 3 was unnecessary (it shows the difference between l1 and l2 regularization), and I thought the pseudo-code in Algorithm 1 was a lot clearer than Figure 4 for showing the scaling factors.
>
> Regarding figure 3, the purpose of this is to give a visualization of the regularization functions to help our non-expert readers with the intuition. We also added a dotted version of the functions to the plot depicting the effect of the scales on the functions. Further, we added a sentence in the body explaining the motivation behind designing it as such (end of pg 4) as well:
>
> “The proposed regularizing terms are inline with the wisdom of the regularization function R(w) = (1 - w^2) \1_{\{|w| \leq 1\}} as introduced in Tang et al. (2017). A primary difference are in introducing a trainable scaling factor, and formulating it such that the gradients capture appropriate sign updates to the weights. Further this regularization term does not penalize weights that are outside of [-1, +1]. One can re-define the function as to include a scaling factor R(w) = (\alpha - w^2) \1_{\{|w| \leq \alpha\}}. In Figure 3, we depict the different regularization terms, to help with intuition”
>
> We removed figure 4 and used the added space to add details to the discussion and experiment section.
> -----------------------------------------------------------------------------------------------------------------
> *comment: Algorithm 1 helped with the clarity of the approach, although it left me with a question: In section 3.3, the authors say that they train by "replacing the sign binarization with the SS_\beta activation" and that they can back-propagate through it. However, in the psuedo-code it seems like they indeed use the sign-function in the forward-pass, replacing it with the signswish in the backward pass. Which is it?
>
> Thank you for pointing out this issue. The pseudo-code is the correct one. We use the derivative of SwishSign function as the approximation in the backward pass as opposed to the straight through estimator. We corrected the sentence in the text pg.5:
>
> “Combining both the regularization and activation ideas, we modify the training procedure by replacing the sign backward approximation with that of the derivative of SS_B activation (\ref{sswish}).”
> ----------------------------------------------------------------------------------------------------------------
>
> *comment: “They modify it by centering it and taking the derivative. I'm not sure I understand the intuition behind using the derivative of the Swish as the new activation. It's also unclear how much of BNN+'s success is due to the modification of the Swish function over using the original Swish activation. For this reason I would've liked to see results with just fitting the Swish.”
>
> The activation function of a BNN is the sign function. The original swish function resembles that of Relu and is not a valid approximation for the sign function as it does not saturate on the right side hence we did not experiment with this function. In training binary networks, there is a discrepancy between the forward pass and backward pass. With the modifications made to the Swish, we attempt to close this gap. One interesting property of this function is it captures gradients over a larger domain as opposed to the straight through estimator (STE) where it immediately reaches zero.
>
> In our experiments we compared with the htan backward approximation, which is the valid alternative that is being used in the literature.
> ----------------------------------------------------------------------------------------------------------------
> *comment: “The fact that some of the architectures did not converge is a bit concerning. It's an important detail if a training method is unstable, so I would've liked to see more discussion of this instability.”
>
> Table 2 is a computationally time consuming task, as we are training the networks on the ImageNet dataset, with limited resources on two different architectures. The total number of experiments are 20. As a result, for those two specific table values our experiments had not terminated by the submission deadline. This sentence hadn’t been worded appropriately. Also we would like to point that we have not experienced any convergence issues with the method.
> ----------------------------------------------------------------------------------------------------------------
> *comment: “The bolding on Table 2 is misleading”
>
> We removed the boldings from table 2 and instead emphasize them in table 3.

---

### Official Review · AnonReviewer3 · 2018-11-04
**Good Paper, which achieves the competitive results over the state-of-the-art methods.**

**Rating:** 8
**Confidence:** 4

**Review:**

1. The abstract of this paper should be further refined. I could not find the technical contributions of the proposed method in it.

2. The proposed method for training BNNs in Section 3 is designed by combining or modifying some existing techniques, such as regularized training and approximated gradient. Thus, the novelty of this paper is somewhat weak.

3. Fcn.3 is a complex function for deep neural networks, which integrates three terms of x. I am worried about the convergence of the proposed method.

4. Fortunately, the performance of the proposed method is very promising, especially the results on the Imagenet, which achieves the highest accuracy over the state-of-the-art methods. Considering that the difficulty for training BNNs, I vote it for acceptance.

---------------------------------

After reading the responses from authors, I have clearer noticed some important contributions in the proposed methods:

1) A novel regularization function with a scaling factor was introduced for improving the capability of binary neural networks;
2) The proposed activation function can enhance the training procedure of BNNs effectively;
3) Binary networks trained using the proposed method achieved the highest performance over the state-of-the-art methods.

Thus, I think this is a nice work for improving performance of  binary neural networks, and some of techniques in this paper can be elegantly applied into any other approaches such as binary dictionary learning and binary projections. Therefore, I have increased my score.

---

> ### Author Response · Authors · 2018-11-20
> **To Reviewer #1**
>
> Thank you for reviewing the paper and providing us comments. Below is a point-point response
> -----------------------------------------------------------------------------------------------------------------
>  *comment: "The abstract of this paper should be further refined. I could not find the technical contributions of the proposed method in it."
>
> We have refined the abstract to include our contributions, please see the revision.
>
> “Deep neural networks (DNN) are widely used in many applications. However, their deployment on edge devices has been difficult because they are resource hungry. Binary neural networks (BNN) help to alleviate the prohibitive resource requirements of DNN, where both activations and weights are limited to 1-bit. We propose an improved binary training method (BNN+), by introducing a regularization function that encourages training weights around binary values. In addition to this, to enhance model performance we add trainable scaling factors in our regularization functions. Furthermore, we use an improved approximation of the derivative of the $\sign$ activation function in the backward computation. These additions are based on linear operations that are easily implementable into the binary training framework and we show experimental results on CIFAR-10 obtaining an accuracy of 86.5%, on AlexNet and 91.3% with VGG network. On ImageNet, our method also outperforms the traditional BNN method and XNOR-net, using AlexNet by a margin of 4% and 2% top-1 accuracy respectively.”
> -----------------------------------------------------------------------------------------------------------------
>
> *comment: "The proposed method for training BNNs in Section 3 is designed by combining or modifying some existing techniques, such as regularized training and approximated gradient. Thus, the novelty of this paper is somewhat weak."
>
> Main contributions of this paper are as follows:
> Suggesting regularization functions that encourage training binary weights
> Embedding trainable scaling factors in the regularization function
> Adaptive backward approximation to the sign derivative
> Admittedly, the notion of regularization as well as that of approximation of the derivative are not new in the literature of binary neural networks (BNN). But, the regularization introduced until then is different from ours because it does not penalize the weights greater than 1 or smaller than -1, also the fact that it is a quadratic function (1 - w^2 in [1]) and does not include a scaling factor. For the gradient approximation,  to the best of our knowledge, there is no adaptive function capable of approximating the derivative of the sign function. Indeed, STE and the one proposed in [2] are both arbitrarily chosen and are fixed approximations. Thus, our novelty is the introduction of the scaling factor into a regularization function constructed for a BNN (class of regularization functions R(w) = |scaling_factor - |weights| |^p where p=1 and 2 in the paper) , as well as an adaptive approximation (using a parameter) of the derivative of the sign function. Hence, using back-propagation, the binary weights and scaling factors are learned using only one objective function.
>
> -----------------------------------------------------------------------------------------------------------------
> *comment: "Fcn.3 is a complex function for deep neural networks, which integrates three terms of x. I am worried about the convergence of the proposed method."
>
> The function can be simplified to take on this form: tanh(beta x / 2) + (beta x / 2) sech^2(beta x / 2). We formulated it as such so that the correspondence with the derivative of the swish function is more clear. Though this the similar complexity as the swish function and as demonstrated by the swish paper [3], as well as our empirical results, we have not observed problems with convergence using the proposed method
>
> [1]Tang, Wei, Gang Hua, and Liang Wang. "How to train a compact binary neural network with high accuracy?." AAAI. 2017.
> [2] Zechun Liu, Baoyuan Wu, Wenhan Luo, Xin Yang, Wei Liu, Kwang-Ting Cheng. ”Bi-Real Net: Enhancing the Performance of 1-bit CNNs With Improved Representational Capability and Advanced Training Algorithm” ECCV. 2018
> [3] Prajit Ramachandran, Barret Zoph, Quoc V. Le. “Searching for Activation Functions.” https://arxiv.org/abs/1710.05941. 2018

---

### Public Comment · (anonymous) · 2018-10-06
**Some questions**

1. As said in the paper, bi-real net uses pretrained network weights as an initialization, then what kind of weight initialization do you use?

2. What's the meaning of "we modify the training procedure by replacing the sign binarization with the SSβ activation (2). During training, the real weights are no longer clipped as in BNN training" in section 3.3? Do the float values generated by the SSβ activation replace the {+1, -1} in forward time? If so, how can you make use of bit-wise operation, which is the key to speed up bnn?

---

> ### Author Response · Authors · 2018-10-09
> **Calrification**
>
> 1. For Cifar10 results the weights were initialized with the Xavier Glorot [1], as for ImageNet the same approach as bi-realnet was used, where a pre-trained full precision network using htan activation was used.
>
> 2. By this we mean, we replace the STE estimator in the backward pass with the derivative of the SS function. The weights are no longer clipped to allow them to move beyond -1, 1 as we factor out a scale for each filter. Lastly, we still do the forward pass using binary values {-1, +1}
>
> [1] Glorot, Xavier, and Yoshua Bengio. "Understanding the difficulty of training deep feedforward neural networks." Proceedings of the thirteenth international conference on artificial intelligence and statistics. 2010.

---

> > ### Public Comment · (anonymous) · 2018-10-17
> > **Forward pass**
> >
> > Thanks for your reply, so in the forward pass, as in other BNNs, the sign function is used, and in the backward pass, the derivative of the SS function is used instead of STE, isn't it? Thanks

---

> > > ### Author Response · Authors · 2018-10-17
> > > **Forward pass**
> > >
> > > Yes, that is correct. You could also refer to the algorithm section in the paper as well.

---

### Public Comment · (anonymous) · 2018-10-07
**How many layers did you use in AlexNet and VGG?**

1.How many layers did you use in AlexNet and VGG?    Did you binarize all layers?

2.As you said   bi-real net introduce a shortcut connection .Did you binarize the shortcut ? You didn't binarize the first layer,how about the last layer?

---

> ### Author Response · Authors · 2018-10-09
> **Architectures**
>
> 1. VGG-16 was used without the two first convolutional layers (conv2d-64 and conv2d64).
>
> conv 256 - conv 256 - conv 512 - conv 512 - conv 1024 - conv 1024 - FC 1024 - FC 1024 - FC (num classes)
>
> For AlexNet:
>
> conv 64 - conv 192 - conv 384 - conv 256 - conv 256 - FC 4096 - FC 4096 - FC (num classes)
>
> Additionally, a BatchNorm layer was added before every activation. In the case of Cifar10 only the first layer was not binarized, whereas in ImageNet both the first layer and last layers were not binarized.
>
>
> 2. The shortcut was not binarized in the results presented in the paper. Also the last layer was not binarized for ResNet-18.

---

> > ### Public Comment · (anonymous) · 2018-10-16
> > **The architectures of AlexNet  used for ImageNet ？**
> >
> > I know your alexnet used for cifar10. What‘s the the architectures of AlexNet  used for ImageNet? can you put up the  framework？

---

> > > ### Public Comment · (anonymous) · 2018-10-16
> > > **Architectures**
> > >
> > > conv 64 - conv 192 - conv 384 - conv 256 - FC 4096 - FC 4096 - FC 4096
> > >
> > > excuse me,is this the  original AlexNet  layers?

---

> > > > ### Author Response · Authors · 2018-10-16
> > > > **correction**
> > > >
> > > > This is the AlexNet from one weird trick (https://arxiv.org/abs/1404.5997 ), I corrected the original statement.

---

> > > > > ### Public Comment · (anonymous) · 2018-10-17
> > > > > **alexnet Architectures**
> > > > >
> > > > > Are you kidding? The AlexNet from one weird trick is
> > > > > conv 64 - conv 192 - conv 384 -conv 384 - conv 256 - FC 4096 - FC 4096 - FC 1000 .

---

> > > > > > ### Public Comment · (anonymous) · 2018-10-17
> > > > > > **effective communication**
> > > > > >
> > > > > > Just a bystander chiming in to make some comments -- please be respectful to others even if you disagree strongly.
> > > > > >
> > > > > > Or, in the case when you are not sure if you would offend someone in the discussion due to language/culture barriers, it would be better to set that expectation, e.g.:
> > > > > >
> > > > > > "Pardon me if I may sound aggressive / offensive, but ..."
> > > > > >
> > > > > > The whole point of discussion is to increase mutual understanding, isn't it?

---

> > > > > > ### Author Response · Authors · 2018-10-17
> > > > > > **corrected**
> > > > > >
> > > > > > Yes, that is the one used. I had made a mistake in writing it down. Sorry for the confusion.
> > > > > >
> > > > > > I modified it to FC(num classes) for both.

---

> > > ### Author Response · Authors · 2018-10-16
> > > **alexnet imagenet**
> > >
> > > Same as above was used.

---

### Public Comment · (anonymous) · 2018-12-08
**Does the regularization function really work?**

1. Since the paper (Tang, AAAI 2017, how to train a compact binary neural network with high accuracy) introduced regularization function 1-w^2, we have tried this technique to improve our performance of BNN. We plotted the weight distribution, and we found that we can't see two peaks around 1 and -1, we only see one peak around 0, which is the same as other normal L1&L2 regularization functions. Can you produce two peaks around alpha and -alpha? If not, can you explain why regularization function is important, in a more reliable way? In the paper, we only see the assumption, you didn't give any plot like weight distributions.

2. As the regularization function is one of your only two contributions, I hope we can see more valid results and analytical explanations and experiments, so we can reproduce it and improve the performance in other tasks.

3. Bi-real net achieves Top-1 56.4% Top-5 79.5% in ResNet-18(ImageNet dataset) while yours is 53% and 72.6% respectively,  maybe your paper do not achieve highest performance over state-of-the art, and your experiments results may not  be enough. Can you show more network experiments like deeper Resnet-34 50 or even Densenet which LQ-Nets actually did and did it well.

---

> ### Author Response · Authors · 2018-12-11
> **Yes, it works**
>
> -----------------------------------------------------------------------------------------------------------------
> *comment: "Since the paper (Tang, AAAI 2017, how to train a compact binary neural network with high accuracy) introduced regularization function 1-w^2, we have tried this technique to improve our performance of BNN. We plotted the weight distribution, and we found that we can't see two peaks around 1 and -1, we only see one peak around 0, which is the same as other normal L1&L2 regularization functions. Can you produce two peaks around alpha and -alpha? If not, can you explain why regularization function is important, in a more reliable way? In the paper, we only see the assumption, you didn't give any plot like weight distributions."
>
>
> The objective of including this modified regularization function is indeed to gradually encourage the weights around –alpha and +alpha. You should try to implement our proposed regularizers. Of course, while training we can clearly see (visually) that the weights are distributed around two peaks. Note that alpha is also trainable in our framework, therefore, the weight distribution is conditional on alpha’s value.
>
>
> -----------------------------------------------------------------------------------------------------------------
> *comment: As the regularization function is one of your only two contributions, I hope we can see more valid results and analytical explanations and experiments, so we can reproduce it and improve the performance in other tasks.
>
>
> Please read the updated version of the paper. As stipulated in the abstract,  “We propose an improved binary training method (BNN+), by introducing a regularization function that encourages training weights around binary values. In addition to this, to enhance model performance we add trainable scaling factors to our regularization functions. Furthermore, we use an improved approximation of the derivative of the sign activation function in the backward computation.” Not only we propose a new way to training binary networks, we also performed an extensive number of experiments to show that the methodology is actually working.
>
> -----------------------------------------------------------------------------------------------------------------
> *comment: Bi-real net achieves Top-1 56.4% Top-5 79.5% in ResNet-18(ImageNet dataset) while yours is 53% and 72.6% respectively, maybe your paper do not achieve highest performance over state-of-the art, and your experiments results may not be enough. Can you show more network experiments like deeper Resnet-34 50 or even Densenet which LQ-Nets actually did and did it well.
>
> As stated in the  discussion section of the paper (Section 4.3), “We did not compare our network with that of Liu et al. (2018) as they introduce a shortcut connection that proves to help even the full precision network.” Indeed, the results you cite are the ones using a modified real-valued weights shortcut connection that makes Bi-real net and BNN+ not comparable. However, in [1], their implementation of Resnet-18 achieves 45.7% top-1 accuracy at most on ImageNet dataset.
>
> Also, as stated in the conclusion, “For future work, we plan on extending these to efficient models such as CondenseNet (Huang et al., 2018), MobileNets (Howard et al., 2017), MnasNet (Tan et al., 2018) and on object recognition tasks.”
>
>
> [1] Zechun Liu, Baoyuan Wu, Wenhan Luo, Xin Yang, Wei Liu, and Kwang-Ting Cheng. Bi-real net: Enhancing the performance of 1-bit CNNs with improved representational capability and advanced training algorithm. In ECCV, 2018.

---

### Public Comment · (anonymous) · 2019-08-13
**Is our research work aimed at publishing papers or seeking truth?**

regarding the results reported from this paper: https://arxiv.org/pdf/1906.08637.pdf, we may think about the question from the title of this post:  is our research work aimed at publishing papers or seeking the truth?

A frustrating fact is that the most recent proposed optimization/quantization tricks for binary neural networks are not really necessary. However, worse still, most people in academia don't think so because they need these so-called improvements to publish their papers. Papers like ABC-Net, Group-Net they claimed they can achieve near fp accuracy but they don't release their code, thus the improvements they claimed are not necessarily to be achievable by most other people.

The paper from https://arxiv.org/pdf/1906.08637.pdf reports a result on binary resnet18 on imagenet: 54.5%/77.8%,  which doesn't use any training tricks like scaling factor, customized gradient, fine-tuning from fp models, fp short-cut etc.
They just use the simplest sign function, STE, Adam and standard training solution. But higher acc can be achieved. However, such paper is often just ignored by academia, because most people from this community think there is no "novel" idea proposed, although sometimes pointing out the truth and establishing a better standard should be really meaningful for the long-term development.

---

### Meta-Review · Area_Chair1 · 2018-12-17
**incremental contribution**

**Confidence:** 4
**Recommendation:** Reject

**Metareview:**

The paper makes two fairly incremental contributions regarding training binarized neural networks: (1) the swish-based STE, and (2) a regularization that pushes weights to take on values in {-1, +1}. Reviewer1 and reviewer2 both pointed out concerns about the incremental contribution, the thoroughness of the evaluation, the poor clarity and consistency of the writing. Reviewer3 was muted during the discussion. Given the valid concerns from reviewer1/2, this paper is recommended for rejection.